# Trends, district-level variations, and socioeconomic disparities in cesarean section delivery in Bangladesh

Md Nuruzzaman Khan[1,2,3‡]*, Md Badsha Alam[1,3‡], Shimlin Jahan Khanam[1,3], M. Mofizul Islam[4], Md Arif Billah[4,5]

1 Department of Population Science, Jatiya Kabi Kazi Nazrul Islam University, Mymensingh, Bangladesh, 2 Nossal Institute for Global Health, Melbourne School of Population and Global Health, University of Melbourne, Melbourne, Australia, 3 Centre for Health Research and Innovation (CHRI), Dhaka, Bangladesh, 4 Department of Public Health, La Trobe University, Melbourne, Australia, 5 Health System and Population Studies Division, International Centre for Diarrhoeal Disease Research, Bangladesh (ICDDR,B), Mohakhali, Dhaka, Bangladesh

‡ These authors are joint first author on this work.
* mdnuruzzaman.khan@uon.edu.au

## Abstract

### Background

Cesarean section (CS) delivery rates have risen dramatically worldwide, with most countries exceeding the World Health Organization's (WHO) recommended rate of 10–15%. However, disparities exist, with evidence suggesting that socioeconomic disadvantage and geographic location significantly influence CS rates. Despite this, comprehensive estimates, particularly in Bangladesh, remain limited. This study aims to examine trends, district-level variations, and socioeconomic disparities in CS rates in Bangladesh.

### Methods

Data from seven rounds of the Bangladesh Demographic and Health Surveys, conducted between 1999/2000 and 2022, were analyzed. The outcome variable was CS delivery, categorized by mode of delivery and place of delivery. Explanatory variables included districts, wealth quintiles, and socio-demographic characteristics. Descriptive statistics were used to illustrate trends and variations in CS delivery over time in Bangladesh. Multilevel mixed-effects binary logistic regressions were employed to identify the factors associated with CS delivery.

### Results

Between 1999/2000 and 2022, hospital births in Bangladesh increased by 55%, largely driven by a significant rise in CS deliveries, from 32% to 69%. Around 85% of the total CS deliveries occurred in the private healthcare facilities in 2022, a marked

**Data availability statement:** The datasets used and analysed in this study are available from the Measure DHS website: https://dhsprogram. com/data/available-datasets.cfm.

**Funding:** The author(s) received no specific funding for this work.

**Competing interests:** The authors have declared that no competing interests exist.

**Abbreviations:** CS, cesarean section; LMICs, low- and middle-income countries; WHO, world health organization; DHS, demographic health survey; BDHS, Bangladesh demographic health survey; NIPORT, national institute of population research and training; PSU, primary sampling unit; aOR, adjusted odds ratio; cOR, crude odds ratio; CI, confidence interval; NGO, non-government organizations; STROBE, strengthening the reporting of observational studies in epidemiology.

increase from 41.5% in 1999/2000. In contrast, CS delivery rates in government healthcare facilities fell from 53% to 13.6% during the same period. Mothers in border and hilly districts, as well as those in the poorest wealth quintile, reported lower rates of CS delivery compared to their counterparts.

## Conclusion

The uneven distribution of CS delivery across districts and socioeconomic groups highlights the need for a more tailored approach to childbirth. While government efforts to reduce unnecessary CS use have been insufficient, this study suggests that a one-size-fits-all strategy may exacerbate disparities. Instead, the focus should shift from increasing access to ensuring justified and appropriate use of CS, with public healthcare facilities playing a crucial role in providing safe alternatives.

## Introduction

The prevalence of cesarean section (CS) deliveries- a life-saving medical procedure- has risen sharply, increasing from approximately 7% in 1990 to an estimated 21% in 2023 [1,2]. Projections indicate that this rate will increase to 30% by 2030, with the majority of the increase occurring in low- and lower-middle-income countries (LMICs) [2]. Each of these estimates, current and future, exceeds the recommended 10–15% delivery in CS, with over 10% conferring no discernible advantages to maternal and child health at the population level [2–4]. While this rise is attributed to improved surgical safety, evolving medical practices, and the enhancement of socio-economic status, it has triggered discussions about its broader public health implications [5].

On the one hand, CS delivery undeniably enhances maternal and neonatal outcomes in specific scenarios, reducing perinatal mortality and morbidity associated with fetal distress, dystocia, and pre-existing maternal health conditions [6,7]. However, the overuse of CS delivery exposes both mothers and infants to unnecessary surgical risks, including infections, hemorrhage, and enduring complications [8,9]. Moreover, it can potentially disturb the delicate establishment of the newborns' microbiome, influencing future health [10].

Furthermore, this increasing trend of CS delivery often prompts a focus on overuse, overlooking the serious lack of access to CS delivery in many LMICs, such as Somalia, or in areas underserved within countries, highlighting the issue of necessity rather than overuse [11,12]. Additionally, disparities in access to these procedures underscore socio-economic inequalities. Wealthier countries and individuals often exhibit higher rates of use, not necessarily due to a higher incidence of pregnancy complications and other cases where CS delivery is recommended, but rather reflecting disparate needs based on economic status [13–15]. This complexity often creates a dilemma, with the genuine need hidden by overuse, contributing to the rise in maternal and child mortality [15]. Addressing this challenge necessitates segregated data for each country and areas within countries, accompanied by comprehensive lists of factors associated with corresponding CS delivery and how such estimates

relate to maternal and child health. However, this is often lacking in LMICs, where data frequently focuses on country-level estimates and socio-demographic factors [3,15–18].

Bangladesh, an LMIC, records over 4 million annual births, with approximately half delivered through CS, significantly contributing to the global increase in CS delivery [15]. This higher rate of CS is often unjustified, as disadvantaged women—particularly those in rural areas with lower education and socio-economic status—report significantly lower CS utilization, even if they need such service [15]. Conversely, CS delivery is highly concentrated among women with improved socio-economic status and those residing in urban areas, as well as their combinations [3,15–18]. This indicates significant regional variations in CS delivery, supported by existing evidence of differing coverage in terms of rurality, education, and socio-economic status [2,15]. However, this nuanced understanding is frequently overlooked in country-level policies and programs in LMICs, particularly in Bangladesh, which tend to focus on controlling the overuse of CS delivery nationwide through uniform rules and regulations rather than considering regional estimates [15]. This approach is primarily based on available data from published research, where the lowest geographical unit is a region (the first and largest administrative unit of Bangladesh) that comprises several districts (the second administrative unit of Bangladesh) with significant variations within a region in terms of rurality, education, and socio-economic status [3,15–18]. A uniform policy that does not account for the variations may overlook the field-level needs, make the programs less effective, and create a pathway to increase maternal and child mortality [15]. Our study aims to estimate trends, district-level variations, and socio-economic disparities in CS delivery in Bangladesh.

## Methods

### Study design and sampling technique

We conducted a comprehensive analysis using data from seven rounds of the Bangladesh Demographic and Health Survey (BDHS) spanning the years 1999/2000–2022. BDHS is a nationally representative household survey, an integral component of the Demographic and Health Survey Program administered by the USA in 89 LMICs. Financial support for these surveys was provided by the United States Agency for International Development (USAID) and technical assistance was rendered by ICF International in Calverton, USA. The survey sample comprised of households selected through a two-stage stratified random sampling technique. In the initial stage, primary sampling units (PSUs) were chosen across the country, utilizing PSU lists derived from the 2011 national population census conducted by the Bangladesh Bureau of Statistics. Subsequently, a household listing operation was executed, and 30 households for 1999/2000–2017/18 and 45 household for 2022 survey were randomly selected from each PSU in the second stage of sampling. Data collection involved surveying women who met specific inclusion criteria: (i) married women of reproductive age (15–49 years old), and (ii) those who were either usual residents of the selected households or had spent the most recent night at the selected households on the day of the survey. Additionally, data were collected for their partners and children under the age of five. For further details, a comprehensive overview of these surveys can be found in the respective survey reports [19–24].

### Analytical sample

We analyzed data from a total of 32,461 participants derived from the six rounds of the BDHS. Of these participants, 4,214 were from the 1999/2000 BDHS, 4,126 from the 2004 BDHS, 3,589 from the 2007 BDHS, 4,956 from the 2011 BDHS, 4,904 from the 2014 BDHS, 5,266 from 2017/18 BDHS and 5,406 were from the 2022 BDHS. The inclusion criteria for sample selection were as follows: (i) having given birth to at least one child within three years preceding the survey for all survey years, and (ii) providing information on delivery methods and the place of delivery.

### Outcome variable

The outcome variable was childbirth through a CS, categorized dichotomously as either "yes" or "no". Relevant data were collected by posing the question to eligible respondents: "*Was the (name) child delivered by cesarean section, that is,*

*did they cut your belly open to take the baby out?*" The response options were yes or no, and these were the parameters considered in this study. We analyzed these responses to estimate two measures: (i) overall CS delivery rates and (ii) institutional CS delivery rates. In both cases, the numerator was the occurrence of CS delivery. However, the denominator considered for overall CS delivery rates was the total number of births, including both home and healthcare facilities, while for institutional CS delivery rates, it was the total number of births occurring at healthcare facilities.

### Explanatory variables

We considered respondents' geographical and socio-demographic factors as explanatory variables, selected based on a review of previous literature [1,3,12,14–17,25,26]. District (second administrative level) and wealth quintile served as the main explanatory variables in our study. The 2022 BDHS included information on participants under both administrative divisions and districts [25]. The wealth quintile, another explanatory variable, was created based on household assets, such as roof types and ownership of televisions, using principal component analysis. The scores generated through this method were classified into five equal groups with cutoff values at every 0.20, designated as poorest, poorer, middle, richer, and richest. Other explanatory variables included respondents' socio-demographic characteristics, namely the mother's age at birth, education, employment status, the sex of the child, exposure to mass media, place of residence, and regional location (divisions).

### Statistical analysis

We estimated the overall and institutional rates of CS delivery and CS delivery at the facility level (CS delivery rate in government healthcare facilities, private healthcare facilities, and NGO healthcare facilities). Overall CS delivery rates were also explored across districts and wealth quintiles. Moreover, we measured the risk ratio of the CS delivery rate by comparing the risk among the mothers in the lowest and richest wealth quintiles. Excess or deficit use of CS delivery for each of the districts was identified by comparing the prevalence of CS delivery for the most recent survey (i.e., 2022 BDHS) using the ideal rate of 10–15%, as considered by the international healthcare community [25]. Therefore, districts that had a higher prevalence of CS than 15% were identified as "excess", within the range of 10–15% were identified as "within range", and less than 10% were identified as "deficit" categories. We used multilevel mixed-effect binary logistic regressions (eq. 1) to investigate factors associated with overall CS delivery, and CS delivery at the institutional level.

$$\mathrm{p}\left(y_i \mid \mathbf{x}'\mathbf{s}\right) = \log\left[\frac{\pi_{ij}}{(1-\pi_{ij})}\right] = c + \beta_1 x_{1ij} + \beta_2 x_{2ij} + \ldots + \beta_n x_{nij} + \cup_{0i} + \epsilon_{ij}$$

(eq. 1)

where, $\pi_{ij}$ is the probability of CS, $c$ is the intercept, $x'\!s_{ij}$ are the independent variables, $\beta'\!s$ are the effect size of the respective independent variables, $\cup_{0i}$ is the cluster level random error, and $\epsilon_{ij}$ is the household level random error.

The rationale for utilizing multilevel regressions was the hierarchical structure of the BDHSs, where individuals are nested within households, and households are nested within clusters. We adopted a two-level multilevel modeling approach, with households and clusters as the two levels. Additionally, we conducted another multilevel model to explore the year-wise changes in the likelihood of CS delivery, treating clusters as the level. All analyses accounted for the complex survey design using Stata's "*svy*" commands. The entire analysis was conducted using STATA software, while Microsoft Excel was used for constructing graphs, and ArcGIS 10.5 for producing maps. This study adhered to the Strengthening the Reporting of Observational Studies in Epidemiology (STROBE) reporting guidelines.

### Ethical consideration

The data analyzed in this study were obtained from the Demographic and Health Survey Program of the USA. Prior to conducting the survey in Bangladesh, approval was obtained from the institutional review board of the ICF, USA, and subsequently from the National Research Ethics Committee of the Bangladesh Medical Research Council. Informed written

consent was obtained from all individuals involved. We obtained permission to access the data for analytical purposes, and the survey authority provided us with deidentified data. As the study involved secondary data analysis and adhered to the relevant guidelines and regulations, no additional ethical approval was required.

## Results

### Background characteristics of the respondents

A majority of the mothers are within the age range of 20–34 years and have attained either a primary or secondary level of education. Almost two-thirds of the mothers reported that they were not engaged in any formal income-generating activities. Mothers living in rural areas constitute 73–84% of the sample. The detailed background characteristics of the respondents are presented in S1 Table.

### Trends in cesarean section delivery rate from 1990/2000–2022, overall and by place of delivery

The trends in CS delivery rate in Bangladesh from 1999/2000–2022, overall and by place of delivery, are presented in Table 1. The institutional delivery rate increased by almost 55% (from 8.7% to 63.5%) over the specified time frame. Concurrently, overall CS delivery demonstrated a growth of approximately 41.2%, escalating from 2.8% to 44.0%. The predominant factor contributing to the overall increase in CS delivery was institutional deliveries. Nearly 69% of institutional deliveries took place in CS delivery, marking a significant rise from the 32% recorded in 1999/2000. There was a substantial surge in CS delivery in private healthcare facilities. In 2022, approximately 85% of total births in private healthcare facilities were delivered via CS delivery, representing a twofold increase from the 1999/2000 CS rate in private healthcare facilities (41.5%). The CS delivery rate in healthcare facilities operated by NGOs remained mostly unchanged over the years, while the CS delivery rate in government healthcare facilities declined more than threefold, decreasing from 53% in 1999/2000 to 13.6% in 2022.

### Variations in institutional and cesarean section delivery rates across districts and divisions

The district-level variations in CS delivery rates as recorded in the 2022 survey are presented in Table 2, with data for divisional variations across survey years shown in S1–S3 Fig. The last column of Table 2 classifies CS rates as "excess," "deficit," or "within range." The highest overall CS delivery rate was 84% in the Bagerhat district, followed by Bandarban (74.8%), Barguna (72.8%), Barishal (71.8%), Bhola (71.4%), and Bogura (69.3%). Compared to the 10–15% CS delivery rate recommended by the international healthcare community, all districts showed excess use, except Rangamati and Cox's Bazar, where the rates fell within the recommended range. Considering divisional variations, the rise in CS delivery

**Table 1. Trends in institutional delivery and cesarean delivery rates in Bangladesh, 1999/2000 to 2022.**

| Survey years | Delivery rate, % (95% CI) | | Institutional cesarean delivery, % (95% CI) | | | |
|---|---|---|---|---|---|---|
| | Institutional or facility delivery rate | Overall cesarean section delivery rate | Total institutional cesarean section delivery rate | CS delivery rate in government health facilities | CS delivery rate in private health facilities | CS delivery rate in NGO health facilities |
| BDHS 1999/2000 | 8.7 (7.5-10.1) | 2.8 (2.3-3.5) | 32.1 (27.4-37.2) | 53.0 (44.1-61.8) | 41.5 (32.6-51.0) | 5.5 (2.5-11.5) |
| BDHS 2004 | 11.7 (10.3-13.4) | 4.5 (3.7-5.4) | 38.4 (33.5-43.4) | 49.0 (39.5-58.6) | 46.5 (38.0-55.1) | 4.5 (1.4-13.4) |
| BDHS 2007 | 17.2 (15.3-19.4) | 8.9 (7.8-10.3) | 51.9 (46.9-56.8) | 31.3 (25.2-38.1) | 61.8 (54.6-68.4) | 7.0 (4.4-11.0) |
| BDHS 2011 | 28.8 (26.8-31.0) | 17.1 (15.6-18.7) | 59.2 (56.0-62.2) | 29.4 (25.4-33.8) | 67.2 (62.9-71.3) | 3.4 (2.2-5.2) |
| BDHS 2014 | 37.6 (34.8-40.5) | 22.9 (20.9-25.0) | 60.9 (58.0-63.6) | 20.9 (17.6-24.6) | 76.3 (72.4-79.9) | 2.8 (1.8-4.4) |
| BDHS 2017/18 | 49.8 (47.5-52.2) | 32.9 (30.9-34.9) | 66.0 (63.7-68.2) | 15.5 (13.5-17.7) | 79.8 (77.4-82.0) | 4.7 (3.7-6.0) |
| BDHS 2022 | 63.5 (61.2-65.8) | 44.0 (41.8-46.3) | 69.3 (66.9-71.5) | 13.6 (12.0-15.3) | 85.1 (83.2-86.8) | 1.3 (0.9-1.9) |

**Table 2. Overall cesarean section delivery rate and institutional cesarean section delivery rate by districts in Bangladesh; BDHS, 2022.**

| District | Overall caesarean section delivery rate, % (95% CI) | Institutional cesarean section delivery rate, % (95% CI) | Deficit or excess in comparison with WHO's recommendation of 10–15% use of CS delivery |
|---|---|---|---|
| Bagerhat[b] | 62.9 (44.6-78.1) | 86.2 (69.6-94.5) | Excess |
| Bandarban[a] | 23.7 (9.7-47.3) | 30.2 (13.9-53.7) | Excess |
| Barguna[b] | 51.3 (33.4-68.9) | 90.6 (73.0-97.2) | Excess |
| Barishal | 42.7 (31.9-54.3) | 77.6 (65.6-86.2) | Excess |
| Bhola | 22.8 (16.0-31.4) | 51.9 (43.3-60.3) | Excess |
| Bogura | 52.7 (39.1-65.9) | 82.5 (64.5-92.4) | Excess |
| Brahamanbaria | 47.5 (29.6-66.1) | 83.9 (70.0-92.1) | Excess |
| Chandpur | 47.8 (33.7-62.3) | 69.6 (52.3-82.8) | Excess |
| Chittagong | 29.4 (22.8-36.9) | 39.2 (32.3-46.6) | Excess |
| Chuadanga[b] | 77.0 (67.6-84.3) | 80.1 (74.5-84.7) | Excess |
| Cumilla | 50.6 (43.4-57.7) | 76.1 (66.6-83.6) | Excess |
| Cox's Bazar | 10.8 (6.7-17.0) | 20.0 (16.5-24.0) | Within range |
| Dhaka | 56.3 (49.0-63.3) | 73.2 (66.8-78.8) | Excess |
| Dinajpur | 59.7 (46.9-71.3) | 75.9 (63.9-84.8) | Excess |
| Faridpur[b] | 61.7 (49.5-72.6) | 89.1 (55.6-98.2) | Excess |
| Feni | 20.6 (9.7-38.4) | 30.9 (17.5-48.4) | Excess |
| Gaibandha | 21.0 (13.0-32.1) | 52.2 (42.1-62.1) | Excess |
| Gazipur | 45.5 (35.4-56.0) | 68.8 (56.4-79.0) | Excess |
| Gopalganj[a] | 22.1 (8.9-45.1) | 52.4 (29.8-74.0) | Excess |
| Habiganj | 31.3 (19.7-45.8) | 48.0 (33.2-63.2) | Excess |
| Joypurhat[a] | 50.6 (24.0-71.0) | 69.4 (49.2-84.2) | Excess |
| Jamalpur | 24.2 (14.0-38.5) | 61.0 (46.0-74.2) | Excess |
| Jessore | 60.8 (48.1-72.1) | 75.5 (62.2-85.3) | Excess |
| Jhalokati[b] | 39.9 (22.4-60.3) | 76.3 (54.5-89.7) | Excess |
| Jhenaidah | 72.8 (65.2-79.2) | 84.4 (75.8-90.3) | Excess |
| Khagrachhari[a] | 13.6 (8.9-20.2) | 100 | Excess |
| Khulna | 60.4 (52.9-67.5) | 71.4 (62.9-78.6) | Excess |
| Kishoreganj | 36.7 (22.9-53.0) | 67.4 (51.8-80.0) | Excess |
| Kurigram | 36.9 (27.9-47.0) | 74.7 (56.6-86.9) | Excess |
| Kushtia | 74.8 (59.9-85.5) | 84.4 (67.5-93.4) | Excess |
| Lakshmipur | 30.8 (18.0-47.5) | 63.9 (50.4-75.5) | Excess |
| Lalmonirhat[b] | 24.4 (16.7-34.3) | 75.8 (66.1-83.5) | Excess |
| Madaripur[b] | 38.0 (23.3-55.4) | 70.2 (45.9-86.7) | Excess |
| Magura[b] | 54.5 (34.5-73.1) | 81.3 (58.5-93.1) | Excess |
| Manikganj[b] | 69.3 (44.1-86.6) | 86.8 (76.4-93.1) | Excess |
| Meherpur[a] | 71.8 (68.2-75.2) | 75.7 (65.4-83.8) | Excess |
| Maulvibazar | 35.9 (26.4-46.6) | 63.4 (50.2-74.8) | Excess |
| Munshiganj[b] | 61.6 (31.8-84.7) | 86.9 (60.6-96.6) | Excess |
| Mymensingh | 45.7 (36.9-54.9) | 79.0 (70.2-85.8) | Excess |
| Naogaon | 62.7 (45.1-77.4) | 79.9 (63.7-90.0) | Excess |
| Narail[a] | 43.1 (36.0-50.5) | 68.9 (51.5-82.2) | Excess |
| Narayanganj | 58.6 (38.3-76.3) | 82.5 (72.8-89.3) | Excess |
| Narsingdi[b] | 60.4 (39.9-77.8) | 94.8 (70.4-99.3) | Excess |
| Natore[b] | 68.6 (49.0-83.3) | 83.5 (66.0-92.9) | Excess |

*(Continued)*

**Table 2.** (Continued)

| District | Overall caesarean section delivery rate, % (95% CI) | Institutional cesarean section delivery rate, % (95% CI) | Deficit or excess in comparison with WHO's recommendation of 10–15% use of CS delivery |
|---|---|---|---|
| Chapai Nawabganj[b] | 45.7 (30.9-61.3) | 77.4 (57.3-89.7) | Excess |
| Netrakona | 30.7 (20.3-43.5) | 72.8 (57.4-84.2) | Excess |
| Nilphamari | 43.1 (27.3-60.4) | 62.3 (44.1-77.6) | Excess |
| Noakhali | 19.1 (8.9-36.2) | 56.1 (37.7-73.0) | Excess |
| Pabna | 46.2 (31.8-61.4) | 65.5 (47.7-79.8) | Excess |
| Panchagarh[b] | 29.8 (14.6-51.2) | 60.3 (47.2-72.1) | Excess |
| Patuakhali | 39.7 (25.7-55.5) | 74.9 (54.4-88.2) | Excess |
| Pirojpur | 45.6 (28.4-63.8) | 72.8 (58.0-83.9) | Excess |
| Rajshahi | 68.7 (55.4-79.5) | 77.7 (65.1-86.7) | Excess |
| Rajbari[a] | 62.7 (41.2-80.1) | 79.7 (65.7-88.9) | Excess |
| Rangamati[b] | 10.0 (6.2-15.8) | 22.7 (16.1-31.1) | Within range |
| Rangpur | 57.4 (46.4-67.7) | 72.6 (61.7-81.4) | Excess |
| Shariatpur[b] | 30.0 (19.3-43.4) | 61.8 (36.6-82.0) | Excess |
| Satkhira | 69.2 (57.7-78.7) | 87.9 (78.5-93.6) | Excess |
| Sirajganj | 37.8 (28.5-48.2) | 71.2 (54.6-83.5) | Excess |
| Sherpur | 35.5 (23.5-49.7) | 59.8 (47.4-71.1) | Excess |
| Sunamganj | 21.3 (14.0-30.9) | 52.4 (42.3-62.3) | Excess |
| Sylhet | 22.0 (15.8-29.8) | 48.9 (37.8-60.1) | Excess |
| Tangail[b] | 71.4 (50.7-85.8) | 86.0 (72.2-93.5) | Excess |
| Thakurgaon[b] | 45.5 (31.1-60.6) | 73.4 (51.1-87.9) | Excess |

**Notes:** Presented as row percentages. [a] Indicates unreliable estimates due to a very small sample size (n<25) and [b] small sample size (n=25–49), warranting caution in interpretation.

rates across surveys is particularly significant (S1 Fig). Furthermore, institutional CS deliveries were more prevalent in most divisions, with a notable increase over the years (S2 Fig), especially in private healthcare facilities (S3 Fig).

## Socio-economic differentials in cesarean section delivery rates across divisions

The rates of CS delivery across wealth quintiles are presented in Fig 1. Among women in the poorest quintile, the CS delivery rate was 10.4%, while among those in the richest quintile, it was 27.7%. This disparity was evident across divisions, with the poorest quintile, women living in the Rangpur division reported the highest CS delivery usage. In contrast, the proportion of CS delivery among the poorest quintile in Dhaka, Sylhet and Chattogram divisions fell below the 15% threshold, indicating a deficit, while the use of CS delivery in Khulna, Rajshahi, and Sylhet divisions remained within the recommended range. Conversely, for the richest quintile, all divisions except Rangpur exhibited excess CS rates. Substantial variations in the risk ratios of CS delivery between the poorest and richest quintiles were observed across districts, ranging from less than 2.0 (Comilla, Jhenaidah, Chuadanga, Bagerhat, Barguna, Barishal, Bhola, Bogura, Brahamanbaria, Chandpur, Cox's Bazar, Dinajpur, Faridpur, Gaibandha, Habiganj, Jessore, Joypurhat and Jhalokati) to 10.8 (in Thakurgaon) (Fig 2). Thus, higher CS delivery rates corresponded to greater relative disparities, with an $R^2$ value of 0.4247.

## Predictors of cesarean section delivery in Bangladesh

Table 3 presents the factors associated with overall and institutional CS delivery, determined using multilevel mixed-effects binary logistic regression models. Mothers of comparatively higher age showed greater likelihoods of CS delivery

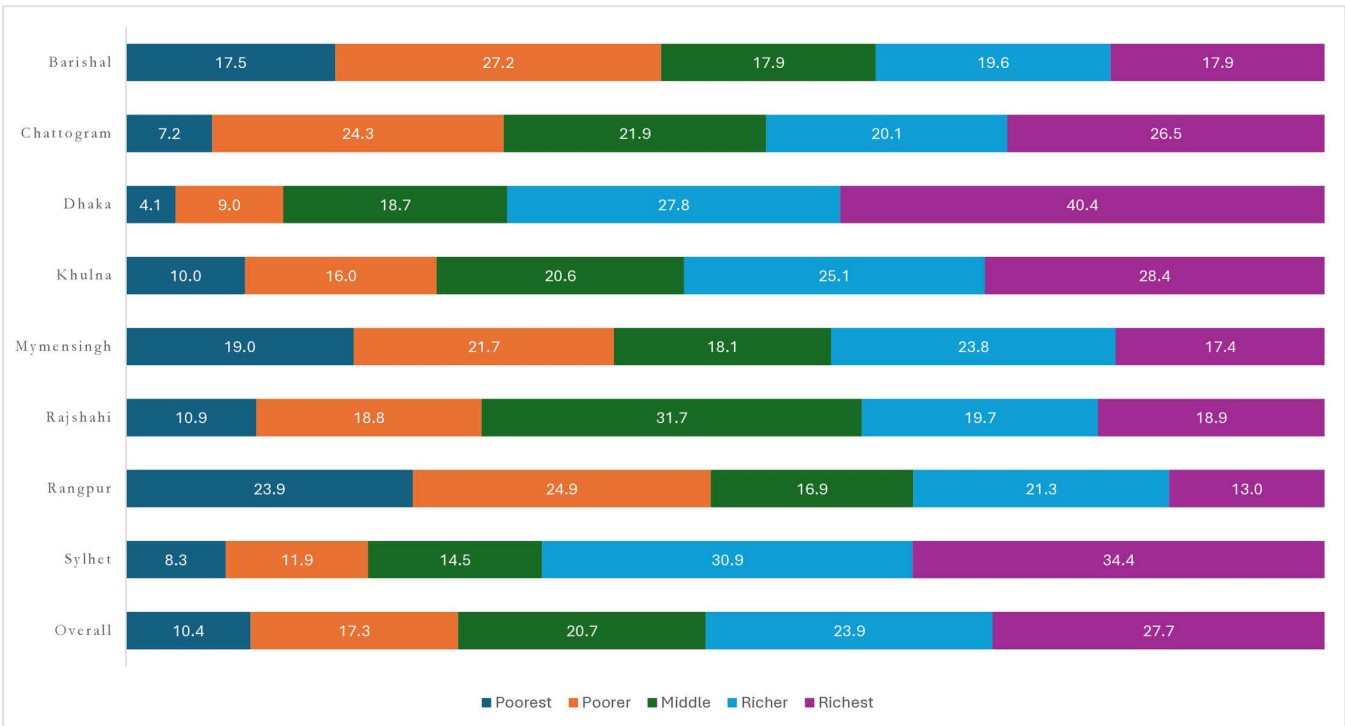

**Fig 1. Cesarean section delivery rates by socioeconomic quintiles across divisions in Bangladesh, BDHS, 2022.**

(aOR: 1.48; 95% CI: 1.04–2.13). Additionally, mothers with higher educational attainment, those in the higher wealth quintile and those not formally employed had higher likelihoods of CS delivery compared to their counterparts. Khulna (aOR: 2.68; 95% CI: 1.77–4.04) and Rajshahi (aOR: 1.69; 1.12–2.56) divisions showed greater likelihoods of CS use compared to mothers in the Barishal division. Conversely, mothers with a parity greater than two and those living in the Sylhet division were less likely to undergo CS delivery. At the institutional level, mothers who were not formally employed and those from higher wealth quintiles reported a higher likelihood of CS delivery. In the pooled regression model (1999/2000–2022), similar trends were observed for both overall CS delivery and institutional CS delivery (S2 Table). In line with the overall increase in CS rate, we found that, compared to 1999/2000, the likelihood of overall CS delivery was 17.86 times higher (95% CI: 14.57–21.89) and institutional CS delivery was 5.34 times higher (95% CI: 4.28–6.77) in 2022.

## Discussion

We observed a concerning trend in CS delivery rates in Bangladesh. From 1999/2000–2022, there was a notable 55% increase in the number of babies born in hospitals. However, a closer examination revealed that CS delivery increased by 41.2% during the same period. Alarmingly, this surge in hospital births was primarily driven by a substantial rise in CS delivery procedures performed in healthcare facilities. In the 1999/2000 survey, 32% of institutional deliveries involved CS delivery, but by 2022, this figure had dramatically risen to 69%. Private healthcare facilities emerged as significant contributors to this increase in CS delivery, accounting for 85% of the country's total CS in 2022, compared to 41.5% in 1999/2000. Conversely, the rate of CS in government healthcare facilities dropped significantly from 53% in 1999/2000 to 13.6% in 2022. The situation becomes even more alarming when examining district-level, divisional-level, and wealth quintile variations, which reveal markedly higher CS delivery rates among selected districts and mothers from higher

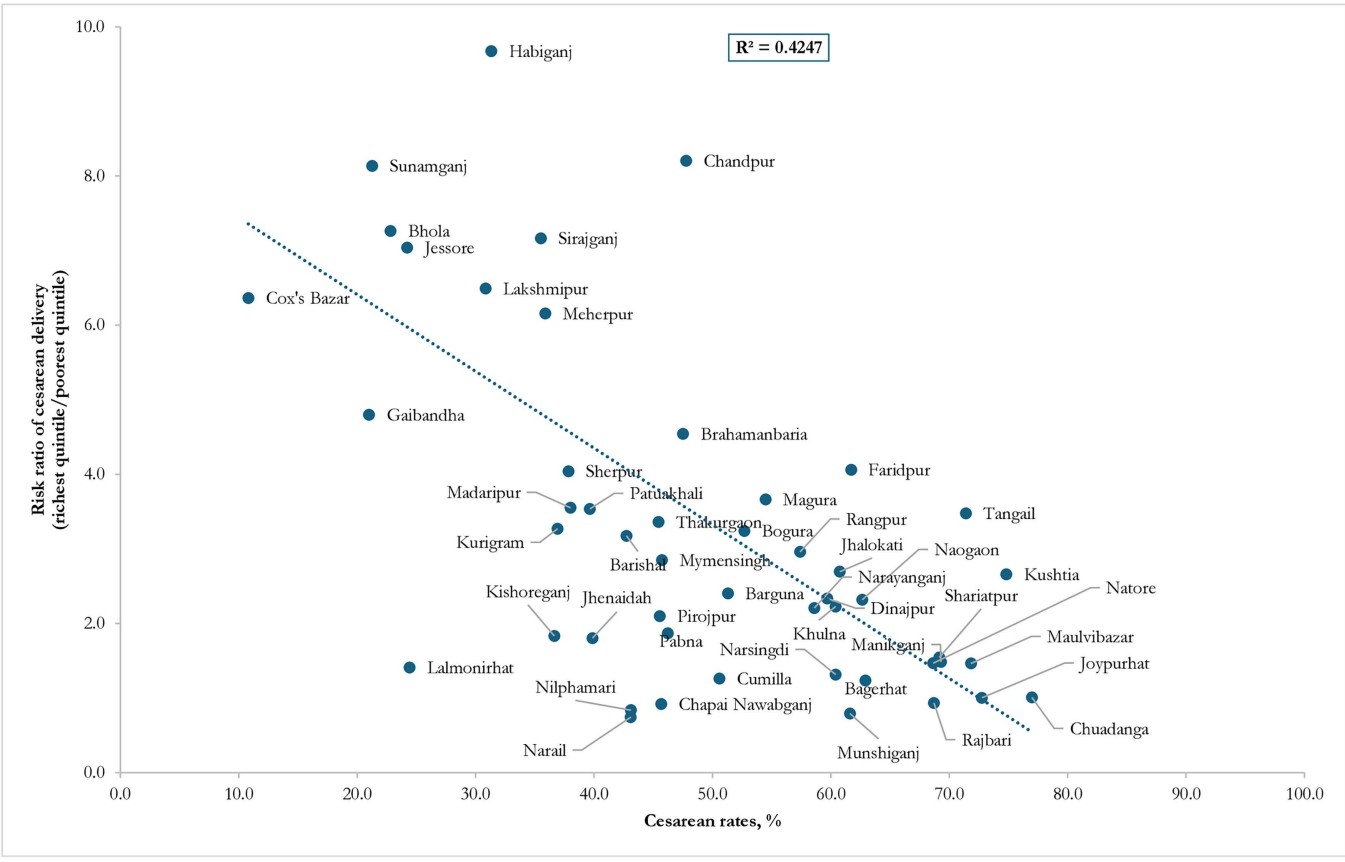

**Fig 2. Cesarean rates and risk ratio of cesarean section deliveries between the poorest and richest quintiles across districts (BDHS 2022).**

wealth quintiles. These findings raise serious concerns and highlight the need for policies and programs aimed at reducing unnecessary CS delivery in Bangladesh.

Since the 2000s, the Bangladesh government has prioritized institutional delivery and access to other maternal health-care services, including antenatal care (ANC) and post-natal care (PNC) [26]. This aligns with Millennium Development Goals (MDGs) and Sustainable Development Goals (SDGs) and was common among LMICs [27]. However, despite significant efforts, progress has been limited. It is also important to acknowledge that ensuring universal coverage of institutional delivery, in line with the SDGs, is unfeasible for Bangladesh, as healthcare facilities are unable to accommodate them all. As demonstrated by this and other studies in Bangladesh, half of all deliveries still occur outside formal healthcare facilities without skilled personnel present [28–31]. This study highlights that the rise in institutional delivery is likely to be driven by an increase in CS use. While we cannot definitively say, without further investigation, what proportion of these CS deliveries are unnecessary, global evidence suggests a substantial proportion is likely to be provider-induced [26,32,33]. Women who choose CS mainly come from affluent wealth quintiles, urban areas, and are educated, or a combination of these factors [15]. This also indicates a pathway where the increasing demand for CS among this advantaged group is driving up its accessibility price, indicating that a segment of women cannot afford these services despite being in greater need due to the ongoing evidence of pregnancy complications in this group where CS delivery is more warranted [15,34]. The very high rate of CS among richer and urban women, along with a deficit of CS delivery among poorer women, supports this conclusion. Encouragement of CS delivery by healthcare facilities is linked with financial

**Table 3. Multilevel mixed-effects logistic regression model assessing the socio-demographic factors associated with cesarean section delivery and cesarean section delivery in institutional level in Bangladesh, BDHS, 2022 (n = 5,405).**

| Characteristics | Overall CS delivery vs overall normal delivery, aOR (95% CI) | Institutional CS delivery vs institutional normal delivery, aOR (95% CI) |
|---|---|---|
| **Mother's age at birth** | | |
| ≤19 years (ref) | 1.00 | 1.00 |
| 20-34 years | 1.28 (1.09-1.50)** | 1.20 (0.97-1.49) |
| ≥35 years | 1.48 (1.04-2.13)** | 1.57 (0.97-2.55) |
| **Mother's education** | | |
| No education (ref) | 1.00 | 1.00 |
| Primary | 1.01 (0.71-1.43) | 1.03 (0.64-1.66) |
| Secondary | 1.53 (1.09-2.15)** | 1.25 (0.79-1.98) |
| Higher | 2.58 (1.77-3.75)*** | 1.49 (0.91-2.44) |
| **Mother's formal employment status** | | |
| Employed (ref) | 1.00 | 1.00 |
| Unemployed | 1.34 (1.13-1.58)*** | 1.24 (1.00-1.54)** |
| **Parity** | | |
| 1-2 (ref) | 1.00 | 1.00 |
| >2 | 0.46 (0.39-0.54)*** | 0.57 (0.46-0.71)*** |
| **Child's gender** | | |
| Male (ref) | 1.00 | 1.00 |
| Female | 0.92 (0.81-1.04) | 1.02 (0.86-1.21) |
| **Wealth quintile** | | |
| Poorest (ref) | 1.00 | 1.00 |
| Poorer | 1.81 (1.46-2.25)*** | 1.68 (1.24-2.26)*** |
| Middle | 2.18 (1.74-2.73)*** | 1.78 (1.32-2.41)*** |
| Richer | 3.08 (2.44-3.89)*** | 2.11 (1.55-2.88)*** |
| Richest | 4.99 (3.83-6.51)*** | 3.00 (2.13-4.23)*** |
| **Place of residence** | | |
| Urban (ref) | 1.00 | 1.00 |
| Rural | 0.94 (0.76-1.16) | 1.17 (0.91-1.52) |
| **Division** | | |
| Barishal (ref) | 1.00 | 1.00 |
| Chattogram | 0.58 (0.40-0.86)*** | 0.39 (0.24-0.64)*** |
| Dhaka | 1.36 (0.93-2.00) | 1.05 (0.64-1.72) |
| Khulna | 2.68 (1.77-4.06)*** | 1.45 (0.85-2.46) |
| Mymensingh | 1.18 (0.77-1.80) | 1.15 (0.66-2.03) |
| Rajshahi | 1.69 (1.12-2.56)*** | 1.24 (0.72-2.13) |
| Rangpur | 1.32 (0.88-1.99) | 0.94 (0.56-1.60) |
| Sylhet | 0.51 (0.32-0.80)** | 0.35 (0.20-0.62)*** |
| **Random effect [a]** | | |
| Cluster level variance (SE)[b] | 0.57 (0.08) | 0.66 (0.11) |
| Intra-class correlation (ICC) | 14.73% | 16.81% |

*(Continued)*

**Table 3.** (Continued)

| Characteristics | Overall CS delivery vs overall normal delivery, aOR (95% CI) | Institutional CS delivery vs institutional normal delivery, aOR (95% CI) |
|---|---|---|
| **Model summary** | | |
| AIC | 6280.25 | 3891.72 |
| BIC | 6425.084 | 4026.63 |

**Notes:** ***p<0.01, **p<0.05, aOR=adjusted odds ratio, CI: Confidence intervals, Ref: Reference group. AIC=Akaike's Information Criterion; BIC=Bayesian Information Criteria; SE=Standard Error. Cluster level variance estimates of random effects are reported. [b] Significance of random effects evaluated by comparing the model with a similar one in which random effects were constrained to zero.

gains rather than its necessity [15]. This is the main reason for the rapid surge of CS delivery in private healthcare facilities over the years, while public healthcare facilities observed a decline in CS, as seen in both this and other studies [15,17,26,28,30,31]. However, whatever the direction, both indicate poor maternal and child health due to the unnecessary use of CS as well as an unmet need for CS [15,17].

Concerns about the rising CS delivery rate in Bangladesh surfaced in the early 2010s. National Safe Motherhood Guidelines and the Maternal Health Voucher Scheme were developed in response, followed by initiatives like enhanced skills training for healthcare professionals, public awareness campaigns, and financial incentives for promoting vaginal delivery [27,35]. However, these programs yielded limited success in curbing the CS delivery surge, coinciding with the increasing role of private healthcare facilities in providing CS delivery [15]. Since 2020, the government has implemented a mandatory CS delivery audit, requiring healthcare facilities to document the reasons for every CS delivery performed [27]. This study reveals significant district-level variations in CS delivery rates. The underlying reasons for these disparities are likely related to healthcare access, socioeconomic conditions, and cultural preferences [15,17,26]. Regions with better healthcare access and higher socioeconomic status tend to have higher CS delivery rates, while rural and economically disadvantaged areas experience lower rates due to accessibility and affordability challenges [15,17]. Community awareness also plays a role, with more informed populations potentially opting for CS delivery based on health considerations [26]. Despite these diverse underlying factors, the overall results suggest that universal policies and programs are a major drawback and indicate the need for more targeted approaches. Policies tailored to specific local needs and contexts could be a more effective way to address the issue of unnecessary CS delivery in Bangladesh. However, the country lags far behind in achieving this target.

The findings of this study have policy implications, particularly concerning the notable variations in CS delivery rates among districts. There are deficits in CS use among women from resource-poor households in many districts, indicating that a one-size-fits-all approach to CS delivery control may not be effective and could potentially worsen maternal and child health outcomes by limiting access for those in genuine need. The recommended approach is to shift focus towards promoting justified CS delivery use, especially in private healthcare facilities where the majority of CS delivery occurs. This entails moving beyond the government's objective of providing CS delivery to enhance child health. The primary goal should be to ensure appropriate and justified CS utilization. In this endeavor, public healthcare facilities, despite their decreasing contribution to overall CS delivery, can play a crucial role. By strengthening their capacity to provide high-quality, justified CS delivery when needed, they can offer a safe and equitable alternative to private facilities, ultimately improving maternal and child health outcomes for all.

## Strengths and limitations

This study's major strength lies in the analysis of seven rounds of nationally representative BDHS data, incorporating large samples. For the first time in Bangladesh, district-level data were analyzed of the BDHS survey, providing estimates for CS delivery and relevant indicators. The analysis also presents CS delivery variations across wealth quintiles,

addressing deficits and excess use of CS delivery. Advanced statistical modeling explored predictors of CS delivery, considering sampling weights in all analyses. These comprehensive analyses provide precise and reliable findings suitable for policy and program development. However, the primary limitation is the use of cross-sectional data, indicating a correlational rather than causal nature of our findings. Data were collected by asking mothers, with limited or no options for validation, which presents a risk of recall bias. Moreover, many factors other than the ones available in the dataset that we considered may influence CS delivery, such as the distance of healthcare facilities. Similarly, the significantly higher rate of CS deliveries in private healthcare facilities warrants further exploration to understand its underlying drivers. However, we were unable to conduct such an analysis due to the absence of relevant variables in the survey data. We also did not present the proportion of total CS deliveries that were medically necessary versus unnecessary, although this distinction is important for interpreting whether the observed trends align with maternal and neonatal health needs.

## Conclusion

The rising trend in CS delivery, particularly in private healthcare facilities, raises critical concerns in Bangladesh. The disproportionate distribution of CS delivery across districts and socioeconomic strata highlights the need for a nuanced approach to childbirth practices. While government efforts to reduce unnecessary CS delivery have had limited success, the study suggests that a one-size-fits-all approach may exacerbate disparities. A shift in focus from merely increasing CS delivery accessibility to ensuring justified and appropriate utilization and proactive role of public healthcare facilities in providing safe alternatives is recommended. Overall, the findings call for a reevaluation of existing regulations and policies to reduce the unnecessary use of CS delivery. Targeted interventions, preferably at the district and lower tiers of local government levels, by appropriate regulatory bodies are needed to ensure transparent, need-based, and accountable cesarean deliveries to improve maternal and child health outcomes.

## Supporting information

**S1 Table. Background characteristics of the study respondents, 1999/2000–2022.**
(DOCX)

**S2 Table. Multilevel mixed-effect logistic regressions assessing the associations between the survey years and overall CS delivery and CS delivery at the institution level in Bangladesh, adjusted for socio-demographic factors; BDHS, 1999/2000–2022 (n = 32,461).**
(DOCX)

**S1 Fig. Trend in cesarean sections delivery rates across divisions in Bangladesh from 1999/2000–2022.**
(DOCX)

**S2 Fig. Trend of cesarean section delivery rates performed in healthcare facility across divisions in Bangladesh from 1999/2000–2022.**
(DOCX)

**S3 Fig. Trend of cesarean section delivery rates performed in private healthcare facility across divisions in Bangladesh from 1999/2000–2022.**
(DOCX)

## Acknowledgments

We extend our gratitude to MEASURE DHS for their valuable data support. Additionally, the authors also acknowledge the support of the Department of Population Science of Jatiya Kabi Kazi Nazrul Islam University, where the study was designed. We also acknowledge to icddr,b, which acknowledges the support of the Government of Bangladesh and

Canada for providing core/unrestricted support for its operations and research. We dedicate this study to the memory of those who fight for an equitable society in Bangladesh.

## Author contributions

**Conceptualization:** Md. Nuruzzaman Khan, Md Badsha Alam, Md Arif Billah.

**Data curation:** Md. Nuruzzaman Khan, Md Badsha Alam.

**Formal analysis:** Md Badsha Alam, Md Arif Billah.

**Software:** Md Badsha Alam.

**Supervision:** Md. Nuruzzaman Khan, M Mofizul Islam.

**Validation:** Md Badsha Alam, Shimlin Jahan Khanam, M Mofizul Islam, Md Arif Billah.

**Visualization:** Md Badsha Alam, Shimlin Jahan Khanam, Md Arif Billah.

**Writing – original draft:** Md. Nuruzzaman Khan.

**Writing – review & editing:** Md. Nuruzzaman Khan, Md Badsha Alam, Shimlin Jahan Khanam, M Mofizul Islam, Md Arif Billah.

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
