## [Decision Letter · Decision Letter 0]

19 Aug 2024

Dear Dr. Khan,

Thank you for submitting your manuscript to PLOS ONE. After careful consideration, we feel that it has merit but does not fully meet PLOS ONE’s publication criteria as it currently stands. Therefore, we invite you to submit a revised version of the manuscript that addresses the points raised during the review process.

We look forward to receiving your revised manuscript.

Kind regards,

Md Mohsan Khudri, Ph.D.

Academic Editor

PLOS ONE

3. In the online submission form you indicate that your data is not available for proprietary reasons and have provided a contact point for accessing this data. Please note that your current contact point is a co-author on this manuscript. According to our Data Policy, the contact point must not be an author on the manuscript and must be an institutional contact, ideally not an individual. Please revise your data statement to a non-author institutional point of contact, such as a data access or ethics committee, and send this to us via return email. Please also include contact information for the third party organization, and please include the full citation of where the data can be found.

5. We note that there is identifying data in the Supporting Information file <file name>. Due to the inclusion of these potentially identifying data, we have removed this file from your file inventory. Prior to sharing human research participant data, authors should consult with an ethics committee to ensure data are shared in accordance with participant consent and all applicable local laws.

-Location data

6. We note that Supplementary figures 2 and 3 in your submission contain [map/satellite] images which may be copyrighted. All PLOS content is published under the Creative Commons Attribution License (CC BY 4.0), which means that the manuscript, images, and Supporting Information files will be freely available online, and any third party is permitted to access, download, copy, distribute, and use these materials in any way, even commercially, with proper attribution. For these reasons, we cannot publish previously copyrighted maps or satellite images created using proprietary data, such as Google software (Google Maps, Street View, and Earth). For more information, see our copyright guidelines: http://journals.plos.org/plosone/s/licenses-and-copyright.

1. You may seek permission from the original copyright holder of Supplementary figures 2 and 3 to publish the content specifically under the CC BY 4.0 license. 

Reviewers' comments:

Reviewer's Responses to Questions

**Comments to the Author**

1. Is the manuscript technically sound, and do the data support the conclusions?

Reviewer #1: Yes

2. Has the statistical analysis been performed appropriately and rigorously?

Reviewer #1: Yes

3. Have the authors made all data underlying the findings in their manuscript fully available?

Reviewer #1: Yes

4. Is the manuscript presented in an intelligible fashion and written in standard English?

Reviewer #1: Yes

Reviewer #1: The authors studied trends, district-level variations, and socioeconomic disparities in cesarean section delivery and its association with neonatal mortality in Bangladesh. Some suggestions were listed as follows:

1. In the Statistical Analysis Section, “We employed multilevel mixed-effect binary logistic regressions to investigate factors associated institutional delivery, CS delivery at the population level, and CS delivery at the institutional level.” Can you provide the statistical equation? How can you explain CS delivery at the population level, since we know that the mixed-effect model is the individual level model?

2. In Table 3, some of the cell frequency is zero; this zero is observed rather than structural; this is observed zero. Could you offer us the confidence interval so we can use small sample inference to estimate the confidence interval?

3. What is the interpretation of R^2=0.2286 in Figure 1?

4. In Table 4 and Table 5, the authors presented the results of mixed effect models. Can you clearly specify what the random effect terms are? Have you only used random intercepts? Could you please provide the value and interpretation of the standard error of random effect terms?

5. How did the mixed effect approaches address changes in CS delivery rate during survey years?

6. How did the mixed effect approaches address changes in neonatal mortality rate during survey years?

**Do you want your identity to be public for this peer review?** For information about this choice, including consent withdrawal, please see our Privacy Policy

Reviewer #1: No

---

## [Author Response · Author response to Decision Letter 1]

18 Oct 2024

We have now added the following sentence with each figure title to address your comment:

The map was created by the authors using a publicly available shapefile from https://data.humdata.org, with ArcGIS 10.1 utilized for this purpose

---

## [Decision Letter · Decision Letter 1]

14 Jan 2025

Dear Dr. Khan,

In BDHS surveys, the sample size was calculated to get reliable estimates at the division level but not at the district level. I wonder how the authors can make sure that the district level estimates are reliable. 

Secondly, how did the authors implement the complex survey design in the multilevel modelling exercise? Please detail out this in the statistical analysis section. 

We look forward to receiving your revised manuscript.

Kind regards,

Mohammed Moinuddin, PhD

Academic Editor

PLOS ONE

Reviewers' comments:

Reviewer's Responses to Questions

**Comments to the Author**

Reviewer #1: All comments have been addressed

Reviewer #2: All comments have been addressed

Reviewer #3: (No Response)

2. Is the manuscript technically sound, and do the data support the conclusions?

Reviewer #1: Yes

Reviewer #2: Yes

Reviewer #3: No

3. Has the statistical analysis been performed appropriately and rigorously?

Reviewer #1: Yes

Reviewer #2: Yes

Reviewer #3: No

4. Have the authors made all data underlying the findings in their manuscript fully available?

Reviewer #1: Yes

Reviewer #2: Yes

Reviewer #3: Yes

5. Is the manuscript presented in an intelligible fashion and written in standard English?

Reviewer #1: Yes

Reviewer #2: Yes

Reviewer #3: No

Reviewer #1: (No Response)

Reviewer #2: Table 2: Quintile analysis would be understandable if calculated with in 100% for all per category.

As this manuscript has data for each district on cs% so low CS rate districts are need to take into consideration for policy making, This should come into conclusion

Just a statement: There are a number of papers published on trends, deteminants, and risk factors using BDHS data. These are already known. Now paper need how to optimize the CS rate in Bangladesh

Reviewer #3: 1) The manuscript partially address the review comments and still require significant improvement in terms of scope of the study, methods, and results interpretation.

2) They estimated proportion of CS for different strata for survey years, for example–Private healthcare facilities, and interpreted as "Private healthcare facilities contributed 84% of the total CS deliveries in 2022, a marked increase from 45.5% in 1999/2000", which is completely wrong interpretation. You cannot estimate contribution of a factor just by estimating prevalence. Be careful about it.

3) The data highlight an overrepresentation of CS delivery in private facilities, but the study does not deeply explore the drivers behind this trend.

4) Although sampling weights were applied, the clustering of data within districts or divisions may introduce sampling bias, especially in areas with smaller sample sizes.

5) The time intervals between BDHS surveys (three to five years) may obscure shorter-term trends or fluctuations in CS delivery rates.

6) The study does not differentiate between medically necessary and unnecessary CS deliveries, which limits the interpretation of whether the observed trends align with maternal and neonatal health needs.

**Do you want your identity to be public for this peer review?** For information about this choice, including consent withdrawal, please see our Privacy Policy

Reviewer #1: **Yes: ** Md. Kamruzzaman, PhD

Reviewer #2: **Yes: ** Aminur Rahman Shaheen

Reviewer #3: No

---

## [Author Response · Author response to Decision Letter 2]

31 May 2025

We have added a MS word file addressing point by point response to each of the reviewers' comments.

---

## [Decision Letter · Decision Letter 2]

6 Oct 2025

Trends, District-Level Variations, and Socioeconomic Disparities in Cesarean Section Delivery in Bangladesh

PONE-D-24-11488R2

Dear Dr. Khan,

We’re pleased to inform you that your manuscript has been judged scientifically suitable for publication and will be formally accepted for publication once it meets all outstanding technical requirements.

Kind regards,

Rajib Chowdhury, M.Sc.; MPH

Academic Editor

PLOS ONE

Additional Editor Comments (optional):

Reviewers' comments:

Reviewer's Responses to Questions

**Comments to the Author**

Reviewer #1: All comments have been addressed

Reviewer #4: (No Response)

2. Is the manuscript technically sound, and do the data support the conclusions?

Reviewer #1: Yes

Reviewer #4: Yes

3. Has the statistical analysis been performed appropriately and rigorously?

Reviewer #1: Yes

Reviewer #4: Yes

4. Have the authors made all data underlying the findings in their manuscript fully available?

Reviewer #1: Yes

Reviewer #4: Yes

5. Is the manuscript presented in an intelligible fashion and written in standard English?

Reviewer #1: Yes

Reviewer #4: Yes

Reviewer #1: All comments have been addressed, and I do not have any further suggestions. Therefore, I believe the manuscript is now ready for acceptance and publication.

Reviewer #4: Dear Authors,

Thank you for carefully addressing the points raised in the previous review. The revised manuscript is now technically sound, clearly written, and entirely consistent with PLOS ONE’s publication criteria. The use of multiple rounds of nationally representative BDHS data, district-level mapping, and multilevel regression provides a comprehensive and credible picture of cesarean section trends, regional heterogeneity, and socioeconomic disparities in Bangladesh.

The revisions have enhanced the clarity of the Methods, the contextualization of the WHO benchmark, and the interpretation of trends between private and government facilities. The supplementary materials are also well organized and support the main findings.

At this stage, I have no further substantive comments to make. Only minor editorial refinements (shortening a few long sentences for flow) might be considered during final preparation, but these are not essential.

Overall, this is an important, rigorous, and policy-relevant contribution to the maternal health literature. I recommend acceptance of this manuscript.

Kind regards,

Sadık Kükrer, M.D., MSc.

**Do you want your identity to be public for this peer review?** For information about this choice, including consent withdrawal, please see our Privacy Policy

Reviewer #1: **Yes: ** Md. Kamruzzaman, PhD

Reviewer #4: **Yes: ** Sadık Kükrer, M.D., MSc.

---

## [Editor Report · Acceptance letter]

PONE-D-24-11488R2

PLOS ONE

Dear Dr. Khan,

I'm pleased to inform you that your manuscript has been deemed suitable for publication in PLOS ONE. Congratulations! Your manuscript is now being handed over to our production team.

Kind regards,

on behalf of

Dr. Rajib Chowdhury

Academic Editor

PLOS ONE